# Ancestral protein sequence reconstruction using a tree-structured Ornstein-Uhlenbeck variational autoencoder

**Lys Sanz Moreta**
Probabilistic programming group
PLTC Section
University of Copenhagen
Copenhagen, Denmark
lys.sanz.moreta@outlook.com

**Ola Rønning & Ahmad Salim Al-Sibahi**
Probabilistic programming group
PLTC Section
University of Copenhagen
Copenhagen, Denmark
{ola,ahmad}@di.ku.dk

**Jotun Hein**
Department of Statistics
University of Oxford
Oxford, United Kingdom
hein@stats.ox.ac.uk

**Douglas Theobald**
Brandeis University
Biochemistry Department
MA, USA
dtheobald@brandeis.edu

**Thomas Hamelryck**
Probabilistic programming group
SCARB / PLTC Section
Departments of Biology / Computer Science
University of Copenhagen
Copenhagen, Denmark
thamelry@bio.ku.dk

## Abstract

We introduce a deep generative model for representation learning of biological sequences that, unlike existing models, explicitly represents the evolutionary process. The model makes use of a tree-structured Ornstein-Uhlenbeck process, obtained from a given phylogenetic tree, as an informative prior for a variational autoencoder. We show the model performs well on the task of ancestral sequence reconstruction of single protein families. Our results and ablation studies indicate that the explicit representation of evolution using a suitable tree-structured prior has the potential to improve representation learning of biological sequences considerably. Finally, we briefly discuss extensions of the model to genomic-scale data sets and the case of a latent phylogenetic tree.

## 1 Introduction

Representation learning of biological sequences is important for data exploration and downstream tasks such as protein design (Detlefsen et al., 2020; Alley et al., 2019). Deep generative models such as variational autoencoders (VAEs) (Kingma & Welling, 2013; 2019) have been especially useful for this purpose (Riesselman et al., 2018; Greener et al., 2018). However, current models do not take evolutionary information fully into account, i.e., by relating the sequences belonging to a protein family in a phylogenetic tree and incorporating parameterized evolutionary models (Durbin et al., 1998). To address this problem, we replace the standard multivariate Gaussian prior of a conventional VAE with a tree-structured prior that takes into account a given evolutionary tree. We propose a prior based on the Ornstein-Uhlenbeck Gaussian process on a tree (Hansen, 1997; Jones &

Moriarty, 2013). We apply the model to a classic problem in phylogenetics, namely the inference of ancestral sequences.

Ancestral sequence reconstruction (ASR), i.e., the inference of ancestral sequences given their descendants or leaf sequences (Pauling et al., 1963; Yang et al., 1995; Koshi & Goldstein, 1996; Joy et al., 2016; Hochberg & Thornton, 2017; Selberg et al., 2021), has important applications including protein engineering (Cole & Gaucher, 2011; Spence et al., 2021), modeling tumour evolution (El-Kebir et al., 2015), evaluating virus diversity and vaccine design (Gaschen et al., 2002), understanding drug mechanisms (Wilson et al., 2015) and reconstructing ancient proteins *in vitro* (Chang et al., 2002; Wilson et al., 2015; Hochberg & Thornton, 2017).

As input, we assume a set of $n_S$ known, aligned leaf sequences and their phylogenetic tree. The task we want to address is the inference of the $n_A \leq n_S - 1$ unknown, ancestral sequences (Joy et al., 2016). We show that our probabilistic model, called Draupnir, is about on par with or better than the accuracy of established ASR methods for a standard experimentally-derived data set (Alieva et al., 2008; Randall et al., 2016) and several simulated data sets. In addition, we show that Draupnir is capable of capturing coevolution among sequence positions, unlike conventional ASR methods.

The paper is organised as follows. In Background, we briefly discuss evolution of biological sequences, ancestral sequence reconstruction and the tree-structured Ornstein-Uhlenbeck process. In Related Work, we discuss deep generative models of biological sequences. In Methods, we describe the Draupnir model, the inference of ancestral sequences and the setup of the benchmarking experiments. In Results, we discuss the quality of the latent representations, compare the accuracy of Draupnir with state-of-the-art phylogenetic methods for ASR, and present the results of ablation experiments. We end with a brief discussion of future work, including extending the method to genomic-scale data sets and the case of a latent phylogenetic tree.

## 2 Background

### 2.1 Protein sequences and evolution

Biological molecules such as proteins and nucleic acids (DNA, RNA) can be characterised by sequences of characters from an alphabet of size $n_C$, where typically $n_C = 21$ for proteins and $n_C = 5$ for nucleic acids (Durbin et al., 1998). These alphabets include one character that represents a gap, which is useful in aligning related sequences in a multiple sequence alignment (MSA). In the course of evolution, mutations arise that cause changes in these sequences, including character substitutions, deletions and insertions. A set of $n_S$ known, homologous, extant sequences and their evolutionary relationships are naturally represented as $n_S$ leaf nodes in a binary tree or phylogeny, where the $n_A$ internal or ancestral nodes represent unknown, ancestral sequences (Joy et al., 2016). Among the internal nodes, the root node is the most ancient node.

Edges between two nodes in the tree are labelled by positive real numbers that represent the time difference or the amount of change between them. Such a labelled binary tree naturally defines a $(n_S + n_A) \times (n_S + n_A)$ matrix containing the pairwise distances between all nodes, called the *patristic distance matrix*, **T**. In the context of biological sequences, the field of phylogenetics is concerned with the inference of the tree topology, the labels of the tree's edges and the composition of the ancestral sequences, making use of methods based on heuristics (such as maximum parsimony) or probabilistic, evolutionary models (Joy et al., 2016).

### 2.2 Ancestral protein reconstruction

The ASR problem amounts to inferring the composition of the $n_A$ ancestral sequences from the $n_S$ extant sequences, making use of a tractable model of evolution (Joy et al., 2016). Typically, the phylogenetic tree that relates the sequences is assumed known. Standard methods to do this typically assume independent (factorized) evolution of the characters

in the sequence, which is a computationally convenient but unrealistic assumption. For example, in proteins, amino acids are involved in an intricate 3-dimensional network of interactions that can lead to strong dependencies between amino acids far part in the sequence. This phenomenon is called *epistasis* (Hochberg & Thornton, 2017), which requires coevolutionary models that go beyond the factorized assumption. Nonetheless, it has been possible to infer ancestral sequences and subsequently resurrect functional ancient, ancestral proteins *in vitro* (Hochberg & Thornton, 2017). The aim of this work is to go beyond the assumption of independent, factorized evolution by using a model of evolution that features continuous, latent vector representations of the protein sequences. This allows us to formulate the ASR problem in the context of a deep generative model.

## 2.3 THE ORNSTEIN-UHLENBECK PROCESS ON A PHYLOGENETIC TREE

Typically, ASR of biological sequences is done using factorised evolutionary models that represent substitutions, insertions and deletions of the discrete characters in the sequences (Joy et al., 2016). In contrast, Draupnir aims to model the evolution of latent, continuous representations or underlying traits of the sequences. A simple diffusive process allowing for an equilibrium distribution is the Ornstein-Uhlenbeck (OU) process (Hansen, 1997; Jones & Moriarty, 2013). As the OU process is a Gaussian process, it has a Gaussian equilibrium distribution, as well as Gaussian marginal distributions.

We use an OU process on a phylogenetic tree (TOU process) (Hansen, 1997; Jones & Moriarty, 2013) to put the latent representations under the control of a parameterized evolutionary model. Apart from the mean, which for our purposes can be assumed to be zero, the TOU process has three parameters: the variation unattributable to the phylogeny or the intensity of specific variation $\sigma_n$, the characteristic length scale of the evolutionary dynamics $\lambda$, and the intensity of inherited variation $\sigma_f$. The covariance function for the corresponding multivariate Gaussian distribution is then given by (Hadjipantelis et al., 2012; Jones & Moriarty, 2013),

$$\Sigma_{k,l} = \sigma_f^2 \exp\left(\frac{-T_{k,l}}{\lambda}\right) + \sigma_n^2 \delta_{k,l} \tag{1}$$

where $T_{k,l}$ is the patristic distance between nodes $k$ and $l$ in the tree, and the Kronecker delta $\delta_{k,l} = 1$ if $k = l$, and 0 otherwise.

The TOU process and related diffusive processes on trees are well-established evolutionary models that have been used to model the evolution of continuous traits, such as body mass or length (Joy et al., 2016). For example, Lartillot (2014) proposes a phylogenetic Kalman filter for ancestral trait reconstruction of low-dimensional, continuous traits; Tolkoff et al. (2018) propose phylogenetic factor analysis, in which a latent variable under the control of a small number independent univariate Brownian diffusion processes is related to observed traits through a loading matrix; Horta et al. (2021) use a multivariate TOU process and Markov chain Monte Carlo to model both continuous traits and sequences of discrete characters. To represent the latter, they make use of a pairwise Potts model.

## 3 RELATED WORK

### 3.1 REPRESENTATION LEARNING OF BIOLOGICAL SEQUENCES

A VAE (Kingma & Welling, 2013; 2019) is a probabilistic, generative model featuring latent vectors or representations, $\{\mathbf{z}\}_{n=1}^N$, that are independently sampled from a prior distribution, $\mathbf{z}_n \sim \pi(\mathbf{z}_n)$. The latent vectors are passed to a neural network (the *decoder*) with parameters $\theta$, leading to a likelihood, $\mathbf{x}_n \sim p_\theta(\mathbf{x}_n \mid \text{NN}_\theta(\mathbf{z}_n))$, for the data, $\{\mathbf{x}\}_{n=1}^N$. The prior is typically a standard multivariate Gaussian distribution, but other priors have been used, such as distributions on the Poincaré ball to recover hierarchical structures (Mathieu et al., 2019). The posterior distribution $p(\mathbf{z}_n \mid \mathbf{x}_n)$ is intractable, but can be approximated with a variational distribution or *guide*, $q_\phi(\mathbf{z}_n \mid \text{NN}_\phi(\mathbf{x}_n))$, involving a second neural network (the *encoder*). Point estimates of the parameters $\theta$ and $\phi$ are obtained by maximizing the

*evidence lower bound* (ELBO),

$$\mathcal{L}_{\theta,\phi}(\mathbf{x}) = \mathbb{E}_q\left[\log\left(\frac{p_\theta(\mathbf{x},\mathbf{z})}{q_\phi(\mathbf{z}\mid\mathbf{x})}\right)\right],$$

using stochastic gradient ascent (Hoffman et al., 2013).

VAEs are increasingly used for representation learning of biological sequences (Detlefsen et al., 2020). Riesselman et al. (2018) use a VAE with biologically motivated priors to evaluate the stability of mutants and to explore new regions of sequence space. Greener et al. (2018) use autoencoders to design metal-binding proteins and novel protein folds. Ding et al. (2019) show that the latent representations obtained with a VAE can capture evolutionary relationships between sequences. The above models do not represent the phylogenetic tree explicitly, but typically aim to condition on some evolutionary information by training on pre-computed MSAs - an approach that has been called *evo-tuning* (Rao et al., 2019; Detlefsen et al., 2020). Hawkins-Hooker et al. (2021) use a VAE with a convolutional encoder and decoder, combining upsampling and autoregression, without relying on a MSA.

The above models assume that the latent vectors factor independently, which is computationally convenient but unrealistic if the sequences are related to each other in a phylogeny. A more realistic approach thus uses a prior $\pi(\{\mathbf{z}\}_{n=1}^N \mid \tau, \kappa)\pi(\kappa)$ that conditions the latent vectors on a given phylogenetic tree, $\tau$, and an evolutionary model with latent parameters, $\kappa$. Because the latent vectors do not factor independently anymore, mini-batch training can include the sequences but not the latent vectors, which limits the possible size of the data sets. Nonetheless, we show here that such a model is both computationally tractable and practically useful for realistic data sets concerning single protein families.

## 4 METHODS

### 4.1 THE DRAUPNIR MODEL

The pseudocode of the Draupnir model is given in Algorithm 1; Figure 1 shows the corresponding graphical model. A summary of the variables, their dimensions and the notation is given in Tables 1 and 2 in Appendix A.1.

As inputs, we assume (a) a set of $n_S$ aligned sequences, each with length $n_L$, organized in the $n_S \times n_L$ matrix, $\mathbf{S}$, and (b) information on their phylogenetic tree in the form of their $(n_S + n_A) \times (n_S + n_A)$ patristic distance matrix, $\mathbf{T}$.

The latent matrix $\mathbf{Z}$ of the model is a matrix with $n_S$ rows (one for each leaf sequence) and $n_Z$ columns, where $n_Z$ is the size of the latent representation of the sequences. In all experiments, $n_Z = 30$. Each column of $\mathbf{Z}$ (with $n_S$ elements) is sampled from a univariate OU process on the phylogenetic tree, representing the evolution of a hidden trait underlying the sequences along the tree. Each row of $\mathbf{Z}$ corresponds to the latent vector of a standard VAE. However, unlike a standard VAE, the latent vectors do not factorize independently.

For each of the $n_Z$ TOU processes, the parameters of the TOU process, corresponding to $\kappa$ in Section 3.1, are sampled from a suitable prior distribution. The TOU process has three parameters: $\sigma_f$, $\lambda$ and $\sigma_n$. As we use $n_Z$ tree OU processes (one for each column of $\mathbf{Z}$), we need to sample $n_Z$ sets of these three parameters.

For each of the three parameters, $\sigma_f, \lambda, \sigma_n$, we sample a hyperparameter $(\alpha_1, \alpha_2, \alpha_3)$ from a half-normal distribution with scale parameter equal to one. These hyperparameters serve as scale parameter for the half-normal priors over $\sigma_f, \lambda$ and $\sigma_n$. Given the parameters of the TOU process obtained from the prior described above, an $n_S \times n_S$ covariance matrix can be calculated based on the patristic distance matrix of the leaves, $\mathbf{T}^{(S,S)}$. We need one such covariance matrix for each of the $n_Z$ columns of the matrix of latent representations, $\mathbf{Z}$. The element $k, l$, with $k, l \in 1, ..., n_S$, of covariance matrix $h$, with $h \in 1, ..., n_Z$, is given by (Hadjipantelis et al., 2012; Jones & Moriarty, 2013),

$$C_{h,k,l} = \sigma_{f,h}^2 \exp(-T_{k,l}/\lambda_h) + \sigma_{n,h}^2 \delta_{k,l}. \tag{2}$$

As decoder, we use a bidirectional gated recurrent unit (GRU, Cho et al. (2014)) with length equal to the alignment length, $n_L$. The input at each position $i$ of the GRU for sequence $\mathbf{S}_{k,:}$ is a concatenated vector, consisting of the latent vector $\mathbf{Z}_{k,:}$ representing sequence $k$, and the BLOSUM embedding $\mathbf{E}_{i,:}$, which is the result of applying a fully connected neural network, $\mathrm{NN}_{\theta}^{(1)}$, to the BLOSUM vector $\mathbf{V}_{i,:}$ describing the amino acid preferences at position $i$ in the MSA (see Section 4.2). For each of the $n_S$ sequences and for each position $i$, the GRU states are mapped to a logit vector that specifies the probabilities of the $n_C$ characters using another fully connected neural network, $\mathrm{NN}_{\theta}^{(2)}$. The architecture of the networks is given in Appendix A.2.

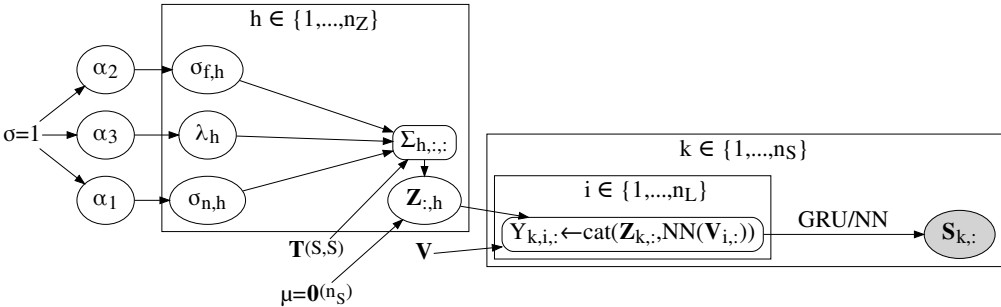

Figure 1: Draupnir as a graphical model. For notation and information on variables and their dimensions see Tables 1 and 2 in Appendix A.1 and A.2. Random variables are shown as ellipses, while deterministic quantities are shown as rounded boxes and observed random variables are shown as shaded ellipses. Parameters of priors and other given quantities are shown without boxes. The model contains three plates, respectively corresponding to the number of dimensions of the leaf sequence-specific latent vector $\mathbf{Z}_{\mathbf{k},:}$ ($n_Z=30$), the number of leaf sequences ($n_S$) and the alignment length ($n_L$). "cat" indicates the concatenation of two vectors. $\alpha$ are the hyperparameters and $\sigma_n, \sigma_f, \lambda$ are the parameters of the TOU processes. $\mathbf{\Sigma}_{h,:,:}$ is the covariance matrix that is used to sample component $h$ of the $n_S$ latent vectors from a multivariate Gaussian distribution (with mean $\mathbf{0}^{(n_S)}$). $\mathbf{T}^{(S,S)}$ is the patristic distance matrix containing the distances between the leaf sequences. $\mathbf{Y}_{k,:}$ is the input vector for the GRU that produces the likelihood parameters for leaf sequence $k$, $\mathbf{S}_{k,:}$. $\mathbf{V}$ represents the MSA as an $n_L \times n_C$ matrix of averaged BLOSUM vectors. NN denotes a fully connected neural network.

## 4.2 BLOSUM EMBEDDINGS

A BLOSUM matrix $\mathbf{B}$ is an $n_C \times n_C$ substitution matrix used for sequence alignment, where each row contains the log-odds scores of replacing a given character with any of the other characters (Henikoff & Henikoff, 1992). Each position (column) in the MSA is represented by the weighted average of the BLOSUM vectors of the characters in that column (see Algorithm 2). The averaged BLOSUM vectors only need to be precomputed once. In the model, the BLOSUM vectors are processed into BLOSUM embeddings by a neural network to provide position-specific information on the MSA, while the latent variables provide sequence-specific information (see Algorithm 2).

## 4.3 MODEL IMPLEMENTATION AND TRAINING

Draupnir was implemented in the deep probabilistic programming language Pyro (Bingham et al., 2019) and trained using stochastic variational inference with Pyro's *AutoDelta* guide by optimizing the ELBO (Kingma & Welling, 2019), resulting in maximum a posteriori (MAP) estimates for all parameters. We use Adam (Kingma & Ba, 2014) as the optimizer using the default values. From the MAP estimates, we can sample the latent representations of the ancestral nodes (see Section 4.4 and equation 5 in the Appendix). These latent representations are then subsequently decoded to their respective ancestral sequences. We

---

**Algorithm 1** The Draupnir model

---

**Require:** Multiple sequence alignment $\mathbf{S}$, patristic distance matrix $\mathbf{T}$

    **for** $j \in [1, 2, 3]$ **do**                        ▷ Hyperpriors over the TOU process parameters
        $\alpha_j \sim \mathcal{HN}(1)$

    **for** $h \in [1, ..., n_Z]$ **do**             ▷ Priors over the parameters of the $n_Z$ TOU processes
        $\sigma_{f,h} \sim \mathcal{HN}(\alpha_0)$
        $\sigma_{n,h} \sim \mathcal{HN}(\alpha_1)$
        $\lambda_h \sim \mathcal{HN}(\alpha_2)$

    **for** $h \in [1, ..., n_Z]$ **do**                   ▷ Kernels for the $n_Z$ TOU processes
        **for** $k, l \in \{1, ..., n_S\}$ **do**
            $C_{h,k,l} \leftarrow \sigma_{f,h}^2 \exp(-T_{k,l}^{(S,S)}/\lambda_h) + \sigma_{n,h}^2 \delta_{k,l}$

    **for** $h \in [1, \ldots, n_Z]$ **do**               ▷ Prior over tree-strucured latent matrix $\mathbf{Z}$
        $\mathbf{Z}_{:,h} \sim \mathcal{MVN}(\mathbf{0}^{(n_S)}, \mathbf{C}_{h,:,:})$

    **for** $i \in [1, \ldots, n_L]$ **do**                       ▷ BLOSUM embeddings
        $\mathbf{E}_{i,:} \leftarrow \mathrm{NN}_\theta^{(1)}(\mathbf{V}_{i,:})$

    **for** $k \in [1, \ldots, n_S]$ **do**                     ▷ Input vector $\mathbf{Y}$ for GRU
        **for** $i \in [1, \ldots, n_L]$ **do**
            $\mathbf{Y}_{k,i,:} \leftarrow \mathrm{cat}(\mathbf{Z}_{k,:}, \mathbf{E}_{i,:})$      ▷ Concatenate sequence- and position-specific vectors

    **for** $k \in [1, \ldots, n_S]$ **do**            ▷ Likelihood parameters (logits) $\mathbf{L}$ from GRU
        $\mathbf{H}_{k,:,:} \leftarrow \mathrm{GRU}_\theta(\mathbf{Y}_{k,:,:})$                  ▷ Bidirectional GRU states
        **for** $i \in [1, \ldots, n_L]$ **do**
            $\mathbf{L}_{k,i,:} \leftarrow \mathrm{NN}_\theta^{(2)}(\mathbf{H}_{k,i,:})$
            $S_{k,i} \sim \mathrm{Categorical}(\mathbf{L}_{k,i,:})$        ▷ Likelihood at position $i$ in sequence $k$

---

**Algorithm 2** Pre-computation of weighted averaged BLOSUM vectors

---

**Require:** Multiple sequence alignment $\mathbf{S}$

    $\mathbf{V} \leftarrow \mathbf{0}^{(n_L \times n_c)}$                       ▷ Initialize BLOSUM weighted average $\mathbf{V}$
    **for** $i \in [1, \ldots, n_L]$ **do**                     ▷ Position in sequence alignment
        **for** $k \in [1, \ldots, n_S]$ **do**                   ▷ Index of leaf sequence
            $r \leftarrow S_{k,i}$                 ▷ Character at position $i$ in leaf sequence $k$
            $\mathbf{V}_{i,:} \leftarrow \mathbf{V}_{i,:} + \mathbf{B}_{r,:}$       ▷ Add BLOSUM vector corresponding to the amino acid
        $\mathbf{V}_{\mathbf{i},:} \leftarrow \frac{1}{n_S} \mathbf{V}_{\mathbf{i},:}$                            ▷ Average

---

also use a custom guide to calculate a variational posterior (Draupnir-variational, see Appendix A.6). Training details can be found in Appendix A.4. All programs were executed on an Intel(R) Xeon(R) Gold 6136 CPU @ 3.00GHz machine with a Quadro RTX 6000 GPU.

## 4.4 INFERENCE OF THE ANCESTRAL SEQUENCES

In this section, for ease of notation, let $\mathbf{z} \equiv \mathbf{Z}_{:,h}^{(A,S)}$ denote one of the $h \in \{1, \ldots, n_z\}$ columns of the latent representation matrix for both ancestral and leaf sequences, $\mathbf{Z}^{(A,S)}$. First, note that $\mathbf{z}$ can be partitioned as $\mathbf{z} = (\mathbf{z}^{(S)}, \mathbf{z}^{(A)})$, where $\mathbf{z}^{(S)}$ and $\mathbf{z}^{(A)}$ denote the latent representations of the leaf and ancestral sequences, respectively. The prior $p(\mathbf{z})$ is a multivariate Gaussian distribution with parameters,

$$\mu = \begin{pmatrix} \mu^{(S)} \\ \mu^{(A)} \end{pmatrix} = \begin{pmatrix} \mathbf{0}^{(n_Z)} \\ \mathbf{0}^{(n_A)} \end{pmatrix}, \mathbf{\Sigma} = \begin{pmatrix} \mathbf{\Sigma}^{(S,S)} & \mathbf{\Sigma}^{(S,A)} \\ \mathbf{\Sigma}^{(A,S)} & \mathbf{\Sigma}^{(A,A)} \end{pmatrix}, \Lambda = \begin{pmatrix} \Lambda^{(S,S)} & \Lambda^{(S,A)} \\ \Lambda^{(A,S)} & \Lambda^{(A,S)} \end{pmatrix},$$

where $\Lambda \equiv \mathbf{\Sigma}^{-1}$. The covariance matrices $\mathbf{\Sigma}^{(S,S)}, \mathbf{\Sigma}^{(A,A)}, \mathbf{\Sigma}^{(A,S)} \equiv \left(\mathbf{\Sigma}^{(S,A)}\right)^T$ are respectively obtained from the distance matrices concerning distances within the leaves, within the ancestors and between the ancestors and the leaves, $\mathbf{T}^{(S,S)}, \mathbf{T}^{(A,A)}$ and $\mathbf{T}^{(A,S)}$, and the TOU process parameters (see Eq. 2).

As $p(\mathbf{z})$ is a multivariate Gaussian distribution, we can easily obtain the conditional distribution of $\mathbf{z}^{(A)}$ given $\mathbf{z}^{(S)}$ as follows (see Bishop (2006), page 689),

$$p(\mathbf{z}) = \mathcal{MVN}\left(\mathbf{z} \mid \mathbf{\Sigma}\right) \Rightarrow p\left(\mathbf{z}^{(A)} \mid \mathbf{z}^{(S)}\right) = \mathcal{MVN}\left(\mathbf{z}^{(A)} \mid \mu_{A|S}, \left(\Lambda^{(A,A)}\right)^{-1}\right),$$

with

$$\mu_{A|S} = \mu_A - \left(\Lambda^{(A,A)}\right)^{-1} \Lambda^{(A,S)} \left(\mathbf{z}^{(S)} - \mu^{(S)}\right) = -\left(\Lambda^{(A,A)}\right)^{-1} \Lambda^{(A,S)} \mathbf{z}^{(S)}.$$

As $\mu_{A|S}$ corresponds to the MAP of $\mathbf{z}^{(A)}$ for any given values of $\mathbf{z}^{(S)}$ and the TOU process parameters when the ancestral sequences are not observed, the MAP estimate of the latent representation of the ancestral sequences is given by,

$$\mathbf{z}^{(A),\text{MAP}} = -\left(\Lambda^{(A,A)}\right)^{-1} \Lambda^{(A,S)} \mathbf{z}^{(S),\text{MAP}}.$$

In addition, a Gaussian approximation of the posterior of $\mathbf{z}^{(A)}$ based on the MAP estimates is given by,

$$\mathbf{z}^{(A)} \sim \mathcal{MVN}\left(\mathbf{z}^{(A)} \mid \mathbf{z}^{(A),\text{MAP}}, \left(\Lambda^{(A,A)}\right)^{-1}\right). \tag{3}$$

The reconstructed ancestral sequences are subsequently obtained by applying the GRU decoder to the MAP estimates of their latent representations, as explained in Section 4.1.

## 4.5 BENCHMARKING

In order to assess the accuracy of the ASR we benchmark Draupnir (MAP and Marginal) against state-of-the-art phylogenetic methods. We selected methods that perform ASR using a given tree topology and given patristic distances, including one Bayesian method (PhyloBayes, Lartillot et al. (2013)) and three maximum likelihood based methods (PAML, Yang (2007); FastML, Ashkenazy et al. (2012); and IQTree, Nguyen et al. (2015)). We apply the ASR methods to both protein sequences, and to their corresponding DNA sequences followed by subsequent translation to protein sequences. For Draupnir, we use the protein sequences, which is the harder problem. We use eleven data sets with different numbers of leaves (from 19 to 800), different alignment lengths (from 63 to 558) and with or without gaps (10 and 1 data set respectively). The data sets include eight simulated data sets generated using the software EvolveAGene (Hall, 2016) and three data sets with experimentally determined ancestral sequences. Note that the simulated data sets were obtained from factored evolution models. We included a large data set with 800 leaves. For prediction with Draupnir, we either use i) the most likely sequence (Draupnir-MAP in Fig. 3), using Equation 4, ii) samples from the marginal distribution (Draupnir-Marginal in Fig. 3; we report the average identity of 50 samples), using Equation 5 or iii) samples from the variational posterior using an amortised guide (Draupnir-Variational, see Appendix A.6), using Equation 6. Details on the data sets and training can be found in Appendix A.3 and A.4, respectively.

## 5 RESULTS

### 5.1 LATENT REPRESENTATIONS AND COEVOLUTION

In order to inspect the quality of the latent representations of the sequences, we use the β-lactamase family with 32 leaf sequences. We visualize t-SNE projections (Van der Maaten & Hinton, 2008) of the latent representations and compare the results with the structure of the phylogenetic tree (see Figure 2). The result indicates that the latent representations represent the structure of the tree and its different clades (subtrees) well, indicating that the TOU process performs well as an informative prior on evolutionary relationships. In Appendix A.6, we show how marginalizing over the latent representations allows Draupnir (marginal and variational) to model coevolution among sites.

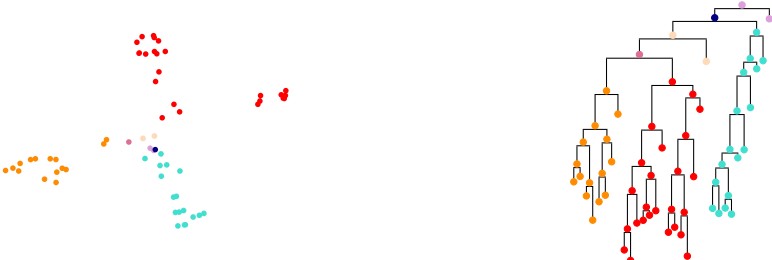

Figure 2: Results for the β-lactamase family with 32 leaves. *Left*: t-SNE projection of the latent representations of the ancestral and leaf nodes. *Right*: The phylogenetic tree. Both plots are coloured according to clade membership.

### 5.2 BENCHMARKING RESULTS

In Figure 3, we compare the accuracy of Draupnir (MAP and marginal) with state-of-the-art ASR methods by plotting the average percent identity between the ancestral sequences as reconstructed by Draupnir and the true sequences. The true ancestral sequences were either experimentally determined or simulated. The description and origin of the data sets can be found in Appendix A.3. Tables with benchmark results are shown in Appendix A.5.

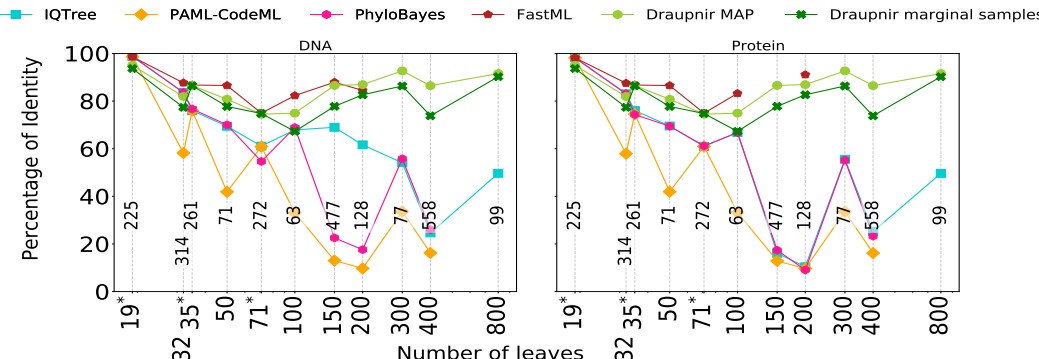

Figure 3: Comparison of the average percentage identity (y-axis) between predicted and true ancestral sequences for Draupnir (MAP and marginal) and ASR methods for data sets with different number of leaves (x-axis; experimental data sets are indicated with an asterisk). Missing points indicate that the ASR method failed to produce results on the given hardware. The alignment size is shown on the dotted lines. We compare with ASR methods using the DNA sequences (*left*) and the corresponding Protein sequences, subsequently translated to protein sequences (*right*). Tables with detailed results for Draupnir-marginal and Draupnir-MAP can be found in Appendix A.5. For the benchmarking settings see Appendix A.8.

## 5.3 ABLATION STUDIES

In the first ablation study, we investigate the influence of the BLOSUM embeddings by removing them as input to the GRU. Overall, the absence of the BLOSUM embeddings slows down convergence and sometimes make the learning process unstable, but ultimately does not strongly affect accuracy (see Figure 5).

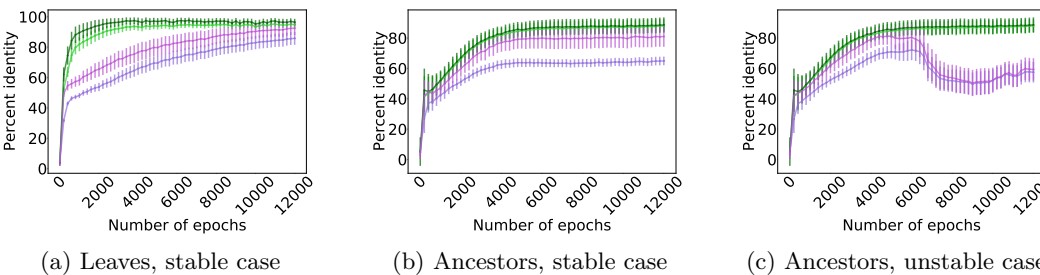

(a) Leaves, stable case     (b) Ancestors, stable case     (c) Ancestors, unstable case.

Figure 5: BLOSUM embedding ablation study for the 800 leaves data set. For every 100 training epochs, the average percent identity and standard deviation are plotted for all leaves (training set) or ancestors (test set), respectively. The results obtained with the BLOSUM embedding are shown in dark green (MAP) and light green (marginal, see Equation 3). The results without the BLOSUM embeddings are shown in pink (MAP) and purple (marginal).

In the second ablation study, we investigate the influence of the tree-structured prior by comparing with a standard VAE with a Gaussian prior. We do this by using diagonal unit covariance matrices for each of the $n_Z$ columns of the latent matrix $\mathbf{Z}$. The rest of the model was identical. We then compare the latent representations obtained for the leaf nodes. The results (see Figure 7) indicate that the standard VAE is not capable of reconstructing the evolutionary relationships well: sequences belonging to the same clade often end up far apart in latent space. This indicates that the influence of the tree-structures prior is substantial. A quantitative analysis of this ablation study can be found in Appendix A.7.

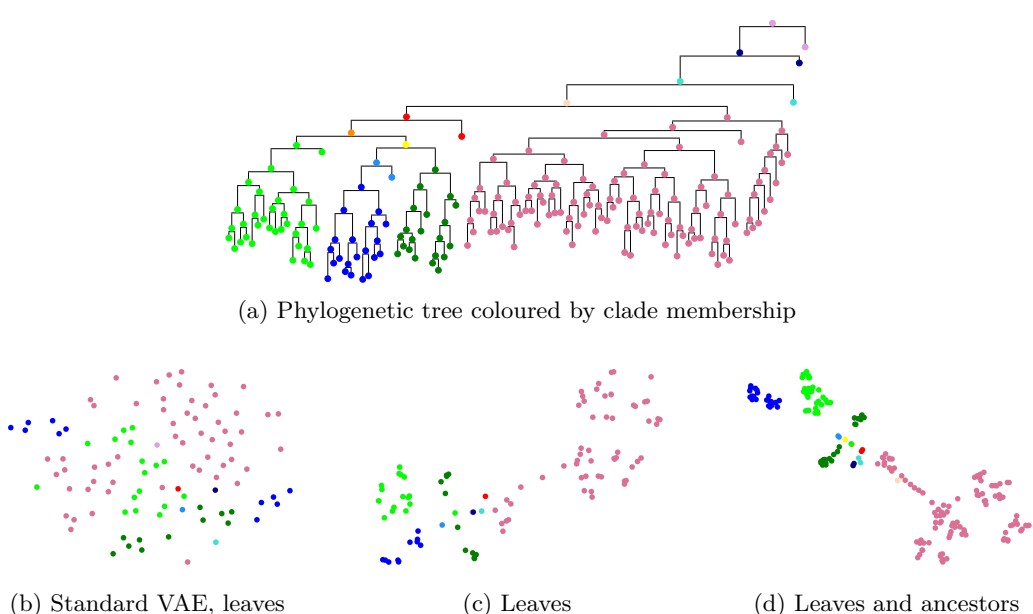

(a) Phylogenetic tree coloured by clade membership

(b) Standard VAE, leaves     (c) Leaves     (d) Leaves and ancestors

Figure 7: Bottom: t-SNE projections of the latent representations for the SRC-Kinase SH3 domain with 100 leaves, obtained from a standard VAE (*left*) and Draupnir-marginal (*center* and *right*), colored by clade. Note that only the latter model can be used to infer the latent representations of the ancestral sequences. *Top*: the corresponding tree.

## 6    DISCUSSION AND FUTURE WORK

Draupnir demonstrates the potential value of incorporating evolutionary information and evolutionary models explicitly in deep generative models for representation learning of biological sequences. We point out that it is possible to extend the model with additional information beyond sequences, for example backbone angles describing protein structure (Golden et al., 2017) or measurements of protein stability. In future work, extending the model to genomic-size data can be done using inducing points for Gaussian processes, as explored in Jazbec et al. (2021) and Vikram et al. (2019). The case of a latent phylogenetic tree can be addressed using a coalescent point process prior (Lambert & Stadler, 2013; Vikram et al., 2019). Finally, in the large data case, the current simple network architectures can be improved with more expressive compositions such as an MSA transformer (Rao et al., 2021) or a deconvolutional model for sequences (Hawkins-Hooker et al., 2021). Finally, we point out that the learned parameters of the TOU processes might offer interpretable information on the evolutionary process.

### ACKNOWLEDGMENTS

*LSM* acknowledges support from the Independent Research Fund Denmark under the grant "Resurrecting ancestral proteins *in silico* to understand how cancer drugs work". *OR* and *ASA* acknowledge support from the Independent Research Fund Denmark under the grant "Deep probabilistic programming for protein structure prediction". We thank Robert Schenck for technical support and contribution of computational resources. We thank the anonymous reviewers for their suggestions and comments.

### AVAILABILITY

Draupnir can be found at `https://github.com/LysSanzMoreta/DRAUPNIR_ASR` and installed as a python library.

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

# A  APPENDIX

## A.1  NOTATION AND VARIABLES

| Name | Description |
|---|---|
| $n_Z$ | Number of TOU processes, length of latent vector (30) |
| $n_L$ | Alignment length |
| $n_S$ | Number of leaf sequences |
| $n_A$ | Number of ancestral sequences ($n_A = n_S - 1$) |
| $n_C$ | Number of character types |
| $\mathbf{S}$ | Sequence alignment matrix of leaf sequences |
| $\mathbf{Z}^{(S)} \equiv \mathbf{Z}$ | Matrix of latent representations of the leaf sequences. Note: We use $\mathbf{Z}$ for notational convenience where possible. |
| $\mathbf{Z}^{(A)}$ | Matrix of latent representations of the ancestral sequences |
| $\mathbf{Z}^{(A,S)}$ | Matrix of latent representations of the leaf and ancestral sequences |
| $\mathbf{T}$ | Patristic distance matrix (given) |
| $\mathbf{T}^{(S,S)}$ | Patristic distance submatrix of distances between leaf sequences (given) |
| $\mathbf{T}^{(A,A)}$ | Patristic distance submatrix of distances between ancestral sequences (given) |
| $\mathbf{T}^{(A,S)}$ | Patristic distance submatrix of distances between leaf and ancestral sequences (given) |
| $\mathbf{B}$ | BLOSUM matrix (given) |
| $\alpha$ | Vector of OU hyperprior parameters |
| $\sigma_{\mathbf{f}}$ | Vector of intensities of inherited variation (TOU process) |
| $\sigma_{\mathbf{n}}$ | Vector of intensities of specific variation (TOU process) |
| $\lambda$ | Vector of characteristic lenght-scales (TOU process) |
| $\mathbf{C}$ | Tensor of $n_Z$ TOU process covariance matrices |
| $\mathbf{V}$ | Matrix of weighted BLOSUM vectors |
| $\mathbf{E}$ | Matrix of BLOSUM embeddings in the model |
| $\mathbf{F}$ | Tensor of BLOSUM embeddings in the guide |
| $\mathbf{Y}$ | Input tensor to GRU |
| $\mathbf{H}$ | State tensor of GRU |
| $\mathbf{L}$ | Tensor of logits of the $n_C$ sequence characters |
| $\theta$ | Parameters of the neural networks and the GRU |
| $\mathbf{0}^{(d)}$ | $d$-dimensional vector of zeros |
| $\mathbf{0}^{(m \times n)}$ | $(m \times n)$-dimensional matrix of zeros |
| GRU | Gated recurrent unit |
| NN | Fully connected neural network |
| $\mathcal{HN}$ | Half-normal distribution |
| $\mathcal{MVN}$ | Multivariate Gaussian distribution |
| cat | Concatenation of two vectors. |

Table 1: Variables and notation used for the Draupnir model.

| Name | Dimensions |
|---|---|
| $\alpha$ | 3 |
| $\sigma_{\mathbf{f}}, \ \sigma_{\mathbf{n}}, \ \lambda$ | $n_Z$ |
| $\mathbf{T}$ | $(n_S + n_A) \times (n_S + n_A)$ |
| $\mathbf{T}^{(A,A)}$ | $n_A \times n_A$ |
| $\mathbf{T}^{(S,S)}$ | $n_S \times n_S$ |
| $\mathbf{T}^{(A,S)}$ | $n_A \times n_S$ |
| $\mathbf{C}$ | $n_Z \times n_S \times n_S$ |
| $\mathbf{Z}^{(S)} \equiv \mathbf{Z}$ | $n_S \times n_Z$ |
| $\mathbf{Z}^{(A)}$ | $n_A \times n_Z$ |
| $\mathbf{Z}^{(A,S)}$ | $(n_S + n_A) \times n_Z$ |
| $\mathbf{Y}$ | $n_S \times n_L \times (n_Z + n_C)$ |
| $\mathbf{E}$ | $n_L \times n_C$ |
| $\mathbf{F}$ | $n_S \times n_L \times n_C$ |
| $\mathbf{H}$ | $n_S \times n_L \times 60$ |
| $\mathbf{L}$ | $n_S \times n_L \times n_C$ |
| $\mathbf{B}$ | $n_C \times n_C$ |
| $\mathbf{S}$ | $n_S \times n_L$ |
| $\mathbf{V}$ | $n_L \times n_C$ |

Table 2: Dimensions of variables

## A.2 DRAUPNIR SETTINGS

**Neural network architecture** The Draupnir model contains three neural networks (see Algorithm 1, Figure 1 and Figure 8): a fully connected network, $\mathrm{NN}_\theta^{(1)}$, that maps the pre-computed BLOSUM vectors, $\mathbf{V}$, to BLOSUM embeddings, $\mathbf{E}$; a bidirectional $\mathrm{GRU}_\theta$ (Cho et al., 2014) with a single layer that takes as input the BLOSUM embeddings and the latent vector of the $k-$th leaf sequence, $\mathbf{Z}_{k,:}$, and a second fully connected network, $\mathrm{NN}_\theta^{(2)}$, that maps the $\mathrm{GRU}_\theta$ states to the logit vectors of the sequence characters. The dimensionality of the state of the bidirectional $\mathrm{GRU}_\theta$ is $2 \times 60$.

In the guide, we re-use the neural network architectures described above: $\mathrm{NN}_\phi^{(2)}$ and $\mathrm{GRU}_\phi$ are identical to $\mathrm{NN}_\theta^{(2)}$ and $\mathrm{GRU}_\theta$, respectively, except that the output size of $\mathrm{NN}_\phi^{(2)}$ is $(2 \times n_Z)$ instead of $n_C$, corresponding to the length of the mean vector and standard deviation vector of the latent representation. $\mathrm{NN}_\phi^{(1)}$ is identical to $\mathrm{NN}_\theta^{(1)}$.

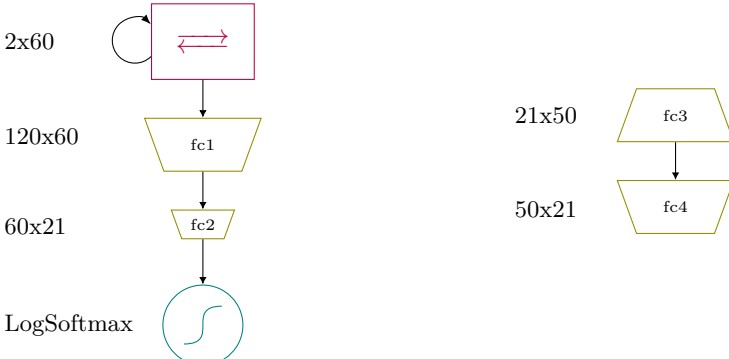

Figure 8: Architectures and dimensions of the neural networks used in Draupnir. *Left*: Architecture of the bidirectional $\mathrm{GRU}_\theta$ (red) and $\mathrm{NN}_\theta^{(2)}$. *Right*: Architecture of $\mathrm{NN}_\theta^{(1)}$. "fc" indicates a fully connected layer.

**Additional Draupnir settings** We chose BLOSUM62 as the base substitution matrix, except for the CFP datasets (71 and 35 leaves) where we use PAM70 due to the presence of special amino acids.

## A.3 DATA SETS

Table 3: Descriptions of the eleven data sets used for benchmarking and the leaves-only data set used in the co-evolutionary analysis. All data sets contain insertions and deletions (gaps), except the one in italic (top), which only contains substitutions. Data sets labelled with an asterisk only contain the sequence of the root node.

| Dataset | Number of leaves | Alignment length | Source |
|---|---|---|---|
| **Datasets with experimentally determined ancestral sequences** | | | |
| *Randall's Coral fluorescent proteins (CFP) benchmark* | 19 | 225 | Randall et al. (2016) |
| *Coral fluorescent proteins (CFP) Faviina subclade | 35 | 261 | *allfav* root node sequence from Alieva et al. (2008) |
| *Coral fluorescent proteins (CFP) subclade | 71 | 272 | *allcor* root node sequence from Alieva et al. (2008) |
| **EvolveAGene4 (Hall, 2005) simulations** | | | |
| Simulation $\beta$-Lactamase | 32 | 314 | GenBank accession no. AF309824 |
| Simulation Calcitonin | 50 | 71 | NBCI CCDS7820.1 |
| Simulation SRC-Kinase SH3 domain | 100 | 63 | GenBank BC011566.1 |
| Simulation Sirtuin | 150 | 477 | NBCI CCDS44412.1 |
| Simulation SRC-Kinase SH3 domain | 200 | 128 | GenBank BC011566.1 |
| Simulation PIGBOS | 300 | 77 | NBCI CCDS81884.1 |
| Simulation Insulin | 400 | 558 | NCBI BC011566.1 |
| Simulation SRC-Kinase SH3 domain | 800 | 99 | GenBank BC011566.1 |
| **Leaves only data set** | | | |
| PF00400 | 125 | 138 | PFAM family no. PF00400 |

## A.4 Training

Table 4: Training settings and running times (Intel(R) Xeon(R) Gold 6136 CPU @ 3.00GHz, Quadro RTX 6000 GPU). On the largest dataset (800 leaves), we make use of a Reduce on plateau learning scheduler combined with plating to further improve the accuracy results.

| Dataset | Epochs | Plate size | Learning rate scheduler | Model parameters | Running times |
|---|---|---|---|---|---|
| **Datasets with experimentally determined ancestral sequences** | | | | | |
| Randall's Coral fluorescent proteins (CFP) benchmark, 19 leaves | 16600 | 19 | No | 52055 | 53 min 39 s |
| Coral fluorescent proteins (CFP) Faviina subclade, 35 leaves | 23000 | 35 | No | 55181 | 1 h 36 min 1 s |
| Coral fluorescent proteins (CFP) clade, 71 leaves | 23000 | 71 | No | 52535 | 1 h 4 min |
| **EvolveAGene4 (Hall, 2005) simulations** | | | | | |
| Simulation $\beta$-Lactamase, 32 leaves | 15000 | 32 | No | 52445 | 1 h 41 min 39 s |
| Simulation Calcitonin, 50 leaves | 18700 | 50 | No | 52985 | 2 h 10 min 1 s |
| Simulation SRC-Kinase SH3 domain, 100 leaves | 21600 | 100 | No | 54485 | 2 h 42 min 10 s |
| Simulation Sirtuin, 150 leaves | 20000 | 150 | No | 55985 | 4 h 6 min 23 s |
| Simulation SRC-Kinase SH3 domain, 200 leaves | 22000 | 200 | No | 57485 | 49 min 21 s |
| Simulation PIGBOS, 300 leaves | 18000 | 300 | No | 60485 | 44 min 1 s |
| Simulation Insulin, 400 leaves | 18400 | 400 | No | 63485 | 3 h 32 min 9 s |
| Simulation SRC-Kinase SH3 domain, 800 leaves | 25000 | 50 | Yes | 141005 | 3 h 39 min 29 s |
| **Leaves only data set** | | | | | |
| PF00400, 125 leaves - Marginal | 23000 | 125 | No | 55235 | 52 min 48 s |
| PF00400, 125 leaves - Variational | 23000 | 125 | No | 137487 | 1 h 21 min 39 s |

## A.5 Benchmarking tables

Table 5: Benchmarking results using protein sequences. The table shows the means and the standard deviation of the percent identity for all the predicted ancestral sequences and their corresponding true sequences. In the case of Draupnir-MAP, the means and standard deviations are calculated using the most likely sequence of each ancestral node. In the case of Draupnir-marginal samples, they are calculated using 50 samples per ancestral node. "Not available" indicates the ASR method failed to produce results for that data set on the given hardware. The results for the standard ASR methods were obtained using the amino acid sequences.

| | Number of leaves | Alignment length | Draupnir MAP | Draupnir marginal samples | PAML-CodeML | PhyloBayes | FastML | IQTree |
|---|---|---|---|---|---|---|---|---|
| Randall's Coral fluorescent proteins (CFP) | 19 | 225 | 95.03±1.29 | 93.67±0.85 | 98.14±1.3 | 98.09±1.03 | 98.17±1.31 | 98.27±1.07 |
| Coral fluorescent proteins (CFP) clade | 71 | 272 | 74.49±1.07 | 74.78±1.09 | 60.96±0.8 | 61.17±0.85 | 74.94±1.07 | 60.96±0.8 |
| Coral fluorescent proteins (CFP) Faviina subclade | 35 | 261 | 86.49±0.98 | 86.46±1.11 | 74.56±1.14 | 74.33±0.86 | 86.76±1.27 | 76.01±1.16 |
| Simulation Beta-Lactamase | 32 | 314 | 81.92±7.97 | 77.37±7.08 | 57.91±22.39 | 83.12±6.13 | 87.52±6.28 | 83.07±6.01 |
| Simulation Calcitonin, 50 | 50 | 71 | 80.77±9.22 | 77.73±7.33 | 41.94±21.21 | 69.45±8.88 | 86.52±6.07 | 69.54±8.83 |
| Simulation SRC-Kinase SH3 domain, 100 | 100 | 63 | 74.94±10.46 | 67.27±8.92 | 33.24±17.33 | 66.78±9.35 | 83.21±8.71 | 66.76±9.17 |
| Simulation Sirtuin SIRT1, 150 | 150 | 477 | 86.59±4.9 | 77.78±3.98 | 12.87±7.16 | 17.36±1.15 | Not available | 16.09±1.36 |
| Simulation SRC-Kinase SH3 domain, 200 | 200 | 128 | 86.92±5.84 | 82.65±4.85 | 9.69±7.99 | 9.10±1.64 | 91.09±4.54 | 10.44±2.71 |
| Simulation PIGB Opposite Strand regulator | 300 | 77 | 92.69±4.08 | 86.34±3.43 | 33.57±10.39 | 55.20±6.12 | Not available | 55.53±5.95 |
| Simulation Insulin Factor like | 400 | 558 | 86.48±4.06 | 73.81±3.16 | 16.17±8.37 | 23.30±1.74 | Not available | 25.22±1.61 |
| Simulation SRC-Kinase SH3 domain, 800 | 800 | 99 | 91.57±4.3 | 90.24±3.63 | Not available | Not available | Not available | 49.63±3.6 |

Table 6: Benchmarking results using DNA sequences. The table is similar to Table 5 but the reconstructions for the standard ASR methods were obtained using DNA instead of amino acid sequences. The DNA sequences were subsequently translated into protein sequences before comparison.

| | Number of leaves | Alignment length | Draupnir MAP | Draupnir marginal samples | PAML-CodeML | PhyloBayes | FastML | IQTree |
|---|---|---|---|---|---|---|---|---|
| Randall's Coral fluorescent proteins (CFP) | 19 | 225 | 95.03±1.29 | 93.67±0.85 | 98.69±0.82 | 98.82±0.83 | 98.59±0.77 | 98.69±0.76 |
| Coral fluorescent proteins (CFP) clade | 71 | 272 | 74.49±1.07 | 74.78±1.09 | 60.88±0.85 | 54.65±0.71 | 74.94±1.07 | 61.10±0.79 |
| Coral fluorescent proteins (CFP) Faviina subclade | 35 | 261 | 86.49±0.98 | 86.46±1.11 | 76.01±1.16 | 76.67±1.13 | 86.76±1.27 | 76.01±1.16 |
| Simulation Beta-Lactamase | 32 | 314 | 81.92±7.97 | 77.37±7.08 | 58.21±22.68 | 83.61±6.15 | 87.66±6.3 | 83.65±6.13 |
| Simulation Calcitonin, 50 | 50 | 71 | 80.77±9.22 | 77.73±7.33 | 41.88±21.14 | 70.01±8.82 | 86.55±6.29 | 69.42±8.89 |
| Simulation SRC-Kinase SH3 domain, 100 | 100 | 63 | 74.94±10.46 | 67.27±8.92 | 33.06±17.56 | 68.90±9.24 | 82.31±9.9 | 67.90±9.31 |
| Simulation Sirtuin SIRT1, 150 | 150 | 477 | 86.59±4.9 | 77.78±3.98 | 13.02±7.25 | 22.54±2.08 | 87.92±8.4 | 68.98±0.79 |
| Simulation SRC-Kinase SH3 domain, 200 | 200 | 128 | 86.92±5.84 | 82.65±4.85 | 9.72±8.17 | 17.60±2.53 | 84.48±8.5 | 61.62±2.71 |
| Simulation PIGB Opposite Strand regulator | 300 | 77 | 92.69±4.08 | 86.34±3.43 | 33.64±10.35 | 55.79±5.23 | Not available | 54.10±5.03 |
| Simulation Insulin Factor like | 400 | 558 | 86.48±4.06 | 73.81±3.16 | 16.23±8.4 | 26.02±2.22 | Not available | 24.63±3.94 |
| Simulation SRC-Kinase SH3 domain, 800 | 800 | 99 | 91.57±4.3 | 90.24±3.63 | Not available | Not available | Not available | 49.66±3.68 |

## A.6 Coevolution analysis

Conventional ASR methods use models that assume factorized evolution (Horta et al., 2021), that is, they assume that each site evolves independently of all other sites in the sequence. In reality, some sites are coupled in evolution, resulting in dependencies between sites. This phenomenon is called epistasis (Hochberg & Thornton, 2017). Draupnir is a model that goes beyond factorized evolution, and thus potentially models coevolving sites. Here, we analyze to what extend this is indeed the case.

In order to evaluate modelling of coevolution, we make use of direct coupling analysis (DCA) (Morcos et al., 2011). DCA identifies coevolving pairs of residues that directly influence each other by calculating a quantity called Direct Information (DI), which is obtained by fitting a Markov random field. We calculated the DI using *ProDy* ((Bakan et al., 2014)). As data set, we used 125 sequences from the WB40 domain of the PF00400 family from the PFAM data base Mistry et al. (2021).

The DIs of the leaf sequences serve as ground truth and are compared with sequences sampled from Draupnir at the root node in three different ways (see below). If coevolution is at least partially modelled, the DIs of the leaf sequences will be similar (but not completely identical) to the DIs of the sampled sequences at the root node. We sample sequences from Draupnir using three different methods.

The first method (MAP) simply uses the MAP estimates of the probability vectors at each position:

$$\mathbf{a} \sim \prod_{i=1}^{n_L} p\left(a_i \mid \boldsymbol{\theta}_i^{(\mathrm{MAP})}\right),$$ (4)

where $\mathbf{a}$ is an ancestral sequence and $\boldsymbol{\theta}_i^{(\mathrm{MAP})}$ is the MAP estimate of the amino acid probability vector at position $i$ for that sequence. This baseline results in independent sites. Picking the most likely amino acid at each position according to the above expression corresponds to Draupnir -MAP in Section 5.2.

The second method (Draupnir-marginal) makes use of the MAP estimates of the leaf representations but marginalizes over the ancestral representations (see Section 4.4):

$$\mathbf{a} \sim \int \left(\prod_{i=1}^{n_L} p\left(a_i \mid \left[\boldsymbol{\theta}\left(\mathbf{z}^{(a)}\right)\right]_i\right)\right) p(\mathbf{z}^{(a)} \mid \mathbf{Z}^{(\mathrm{MAP})}) \mathrm{d}\mathbf{z}^{(a)},$$ (5)

where the probability vectors $\boldsymbol{\theta}_i$ are obtained from a GRU applied to the latent representation $\mathbf{z}^{(a)}$ of the ancestral sequence $\mathbf{a}$ (see Algorithm 1). The last factor involves conditional Gaussian distributions (see Equation 3 and Section 4.4). Integrating over $\mathbf{z}^{(a)}$ introduces correlations along the sequence. Therefore, this method is in principle capable to model some coevolution as we integrate over the latent representations of the ancestors (while using a point estimate for the latent representations of the leaves). This corresponds to the "Draupnir marginal samples" in Section 5.2.

In the third method (Variational), we make use of a guide $q_\phi(\mathbf{Z} \mid \mathbf{S})$ to obtain a variational posterior over the latent representations $\mathbf{Z}$ (see Algorithm 3):

$$\mathbf{a} \sim \int \left(\prod_{i=1}^{n_L} p\left(a_i \mid \left[\boldsymbol{\theta}\left(\mathbf{z}^{(a)}\right)\right]_i\right)\right) p(\mathbf{z}^{(a)} \mid \mathbf{Z}) q_\phi(\mathbf{Z} \mid \mathbf{S}) \mathrm{d}\mathbf{z}^{(a)} \mathrm{d}\mathbf{Z}.$$ (6)

In this case, we marginalize over the latent representations of both leaves and ancestors. This method should capture the coevolutionary signal to a greater extent than the second method.

The results are shown in Figure 9. As expected the third method, (Variational) does best, while the first method (MAP) does worst. The Variational method improves considerably upon the Marginal method, indicating that replacing the MAP estimate with a variational distribution for the latent representations of the leaves has significant impact. The results indicate that Draupnir indeed to a significant extent can capture coevolutionary information.

---

**Algorithm 3** Architecture of the variational guide, $q_\phi(\mathbf{Z} \mid \mathbf{S})$. We use point estimates for the hyperparameters and parameters of the TOU processes, and multivariate diagonal Gaussian distributions for the latent representations. An asterisk indicates the value of a point estimate. $\delta(.)$ is the Dirac delta function.

---

**Require:** Multiple sequence alignment $\mathbf{S}$

    **for** $j \in \{1,2,3\}$ **do**               $\triangleright$ Point estimates of TOU hyperprior parameters
         $\alpha_j \sim \delta(\alpha_j^*)$

    **for** $h \in \{1,...,n_Z\}$ **do**      $\triangleright$ Point estimates of the parameters of the $n_Z$ TOU processes
         $\sigma_{f,h} \sim \delta(\sigma_{f,h}^*)$
         $\sigma_{n,h} \sim \delta(\sigma_{n,h}^*)$
         $\lambda_h \sim \delta(\lambda_h^*)$

    **for** $k \in [1, \ldots, n_S]$ **do**
        **for** $i \in [1, \ldots, n_L]$ **do**
             $r \leftarrow S_{k,i}$             $\triangleright$ Amino acid at position $i$ in leaf sequence $k$
             $\mathbf{F}_{k,i,:} \leftarrow \mathrm{NN}_\phi^{(1)}(\mathbf{B}_{r,:})$          $\triangleright$ BLOSUM embedding

    **for** $k \in [1, \ldots, n_S]$ **do**
         $\mathbf{H}_{k,:,:} \leftarrow \mathrm{GRU}_\phi(\mathbf{F}_{k,:,:})$          $\triangleright$ Bidirectional GRU states
         $\mathbf{m}, \mathbf{c} \leftarrow \mathrm{NN}_\phi^{(2)}(\mathbf{H}_{k,-1,:})$          $\triangleright$ Mean and standard deviations of $\mathbf{Z}_{k,:}$
         $\mathbf{Z}_{k,:} \sim \mathcal{MVN}(\mathbf{m}, \mathrm{diag}(\mathbf{c}))$

---

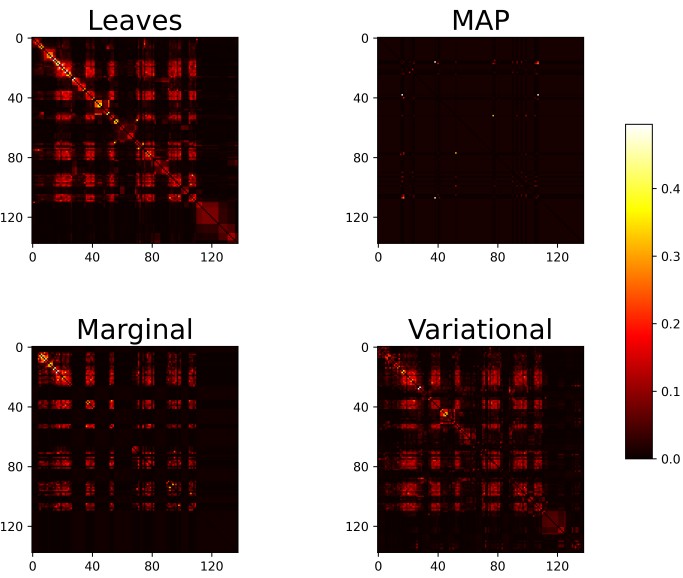

Figure 9: DI values for all position pairs of the WB40 data set obtained from the leaf sequences (upper left) and sequences sampled at the root node (other plots) using the MAP, Marginal and Variational methods. We sampled 125 root sequences, which is equal to the number of leaves. The correlation coefficients between the DIs of the leaves and the sampled sequences are 0.05 (MAP), 0.66 (Marginal) and 0.79 (Variational).

A.7    QUANTITATIVE ANALYSIS OF THE LATENT SPACE REPRESENTATIONS

In order to compare the standard VAE with Draupnir in a quantitative way (see Figure 7), we analyze the correlation between (a) the Euclidean distances between the latent representations of the leaves and (b) the corresponding branch lengths in the phylogenetic tree for both models. The results are shown in Figure 10.

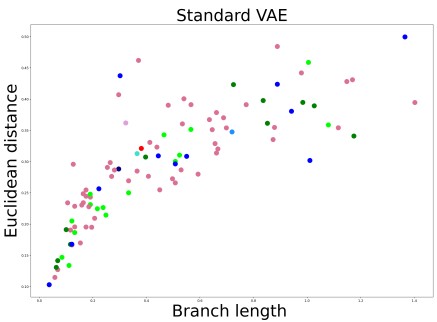 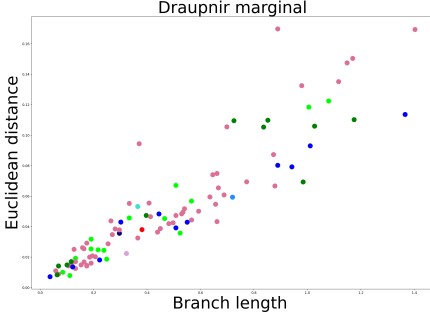

Figure 10: Comparison of the Euclidean distances between the latent representations of the leaves (y-axis) and the corresponding branch lengths in the phylogenetic tree (x-axis). We use the same color scheme as in Figure 2. We traverse the tree in level order and assign the colour of the clade of the first leaf. (*Left*) The standard VAE. The correlation coefficient is 0.79; the Spearman correlation coefficient is 0.85. (*Right*) Draupnir. The correlation coefficient is 0.91; the Spearman correlation coefficient is 0.94.

## A.8    BENCHMARK SETTINGS

**PAML-CodeML settings**    PAML-CodeML (provided by Biopython 1.78) was used with the following settings:

| | |
|---|---|
| verbose | 2 (includes detailed information of the posterior probabilities per node) |
| runmode | 0 (utilize given tree) |
| seqtype | 2 (amino acids) |
| clock | 0 (no molecular clock, genes are evolving at different rates) |
| aaDist | 0 |
| getSE | 0 |
| RateAncestor | 1 |
| aaRateFile | WAG |
| method | 1 |
| model | 2 (more dn/ds (selection coefficient) ratios per branch) |
| fix_alpha | 0 (estimate gamma) |
| alpha | 0.5 (initial alpha value for gamma distribution) |
| fix_blength | 0 (keep given branch lengths) |

**PhyloBayes 4.1 settings**

pd -s -f -T *treefile* -cat -gtr -d *alignmentfile chainname*

In the case of PhyloBayes, as recommended, we run 2 Markov chains until the convergence criteria are met. The recommended convergence criteria are minimum effective sample size above 300 and *max diff* among the chains below 0.1. Both for evaluation of the convergence of the chains and for sampling the ancestral sequences we use 100 samples.

**FastML v3.11 settings**

perl fastml –MSA-file *alignmentfile* –seqType *aa* or *nuc* –SubMatrix *WAG or GTR* –indelReconstruction *ML* –Tree *treefile*

**IQ-Tree v2.0.3 settings**
For IQ-Tree (multicore version 2.0.3), model choice and settings are automatically optimized by using the options "-m TEST" and "-nt AUTO".

iqtree -s *alignmentfile* -m *TEST* -asr -te *treefile* -nt *AUTO*

