# OpenReview forum: "Ancestral protein sequence reconstruction using a tree-structured Ornstein-Uhlenbeck variational autoencoder"
_ICLR.cc/2022/Conference — ICLR 2022 Poster_

### Official Review · Reviewer_mia3 · 2021-10-20

**Correctness:** 3
**Technical Novelty And Significance:** 3
**Empirical Novelty And Significance:** 3
**Recommendation:** 8
**Confidence:** 5

**Main Review:**

Pros:

- I think the authors approach is statistically well grounded and applied to the appropriate problem.

- The results in Figure 3 are quite convincing: This approach is state-of-art for ancestral sequence reconstruction.

- A5 Table 5 and Table 6 are quite convincing. I think it is interesting that some of the other approaches can't even run with the size of the multiple sequence alignment and sequences!

Cons:

- I think there could be better metrics to assess the structure of the latent variables. Could the pairwise distance of latent variables be compared to the distance when fitting a proper phylogenetic tree? Otherwise this comparison is entirely qualitative with tSNE.

Neutral:

- Can the BLOSUM parameters be estimated while fitting the model instead of being precomputed (i.e. they are free parameters)? If there is ablation (Figure 5), does the GRU still learn the covariance of the utilization of amino acids at that position?

- Could you better describe the approach to sample sequences from the model, as discussed in section 4.5? It would be interesting to use something like beam search to generate even better sequences from the model.

- It would be interesting to compare a direct generative approach to trees to a Gaussian Process Variational Autoencoder (https://arxiv.org/abs/1810.11738) with kernels, in which the kernel encodes distance  in a precomputed tree.

- Since these sequences are aligned (thus all the same length), why does a GRU decoder need to be used? Why can't an MLP be used to decode?

**Summary Of The Paper:**

The authors propose replacing the diagonal gaussian prior in a variational autoencoder with an Ornstein-Uhlenbeck process for fitting a generative model to biological sequences. They they apply their approach to ancestral sequence reconstruction.

**Summary Of The Review:**

The authors incorporate biologically-motivated priors into deep models for biological sequences. Their approach is statistically well grounded, and they apply their model to a relevant, difficult task in biology and improve upon it.

---

> ### Author Response · Authors · 2021-11-19
> **Author's comment**
>
> We agree that the article would benefit from more analysis. We have added a quantitative test to the article, concerning the modelling of epistasis / co-evolution (see answer 3 to reviewer 1Min). We have quantified the analysis of Fig. 7, see Appendix A.7.
>
> Estimating the BLOSUM parameters is essentially already incorporated, as the BLOSUM parameters are processed by a neural network with optimizable parameters (see Algorithm 1, Blosum embeddings).
>
> We added more information on sampling from the model (including sampling from a variational posterior using an amortised guide) to the supplementary information.
>
> Comparison with Gaussian process VAEs would indeed be interesting, especially using inducing points to extend the method to the big data case. This is clearly beyond the scope of the rebuttal period. However, we point out that the TOU is itself a Gaussian process that can be interpreted as an evolutionary model.
>
> Despite using aligned sequences, we indeed did not use a MLP, but a GRU (in order to keep the number of parameters under control). Therefore, we have investigated the effect of replacing the GRU with a 5-layer MLP for the smallest and the largest data sets. The accuracy decreases considerably for the large data set, while remaining the same for the small data set.
>
> In order to assess the influence of the GRU on the mapping of the latent space to the sequence, we compared the GRU with a fully connected neural network consisting of five layers with sizes [$n_z,60,120,120,60,n_c$]. The average accuracy of the reconstructed ancestral sequences for two data sets is shown in the table below. For the large data set, the GRU performs considerably better.
>
>
> |                    Dataset                   |       GRU      |      5-MLP     |
> |:--------------------------------------------:|:--------------:|:--------------:|
> | Randall's Coral fluorescent proteins (CFP)   | $94.61\pm1.59$ | $93.15\pm2.08$ |
> | Simulation SRC-kinase SH3 domain, 800 leaves | $88.62\pm4.97$ | $62.39\pm3.61$ |

---

### Official Review · Reviewer_Exob · 2021-11-01

**Correctness:** 3
**Technical Novelty And Significance:** 3
**Empirical Novelty And Significance:** 1
**Recommendation:** 5
**Confidence:** 4

**Main Review:**

positives:

1) The motivation of the paper is clear and interesting: the standard methods for phylogenetic inference typically assume independent evolution of the characters in the sequence and the paper proposes to use VAEs in combination with the OU process in the latent dimension to capture possible higher order interactions (epistatsis) in the sequence evolution using a (given) tree structure.

2) The description of the steps and derivations are clear and the approach seems novel and interesting.

negatives:

1) However, the paper falls short in the results section demonstrating the claims and power of the proposed method: It is hard to see the improved performance in terms of accuracy of the method from Figures 2/3/4 given the gap. Adding some standard deviation bars will help a lot in demonstrating this claim. Also seems that the method is mostly outperformed by FastM: the paper would benefit from a discussion on the method of choice for small sequences; is there any reason to pick the proposed method over AutoML when both run.

2) A very important ablation study has been done in Figure 7 to asses the importance of the added tree structure in the latent space, however, the result is purely quantitative and relies on TSNE embedding. A more quantitive analysis similar to Figure 3 would improve the paper.

4) The paper alludes to computational benefits of the method but never completely delve into the computational aspect of the work. If the computational side is really the selling point I suggest adding plots on the time complexity and discussion on why the propsoed method achieves a better scaling compared to others.

5) I think beyond the usual metrics (accuracy, scaling, and time), the paper can investigate more on its strength which is the ability to generate samples informed by the evolutionary process. Therefore, it can be the case that the latent dimension learned in the proposed VAEs provide some evolutionary information beyond the usual phylogenetic trees. An interesting one that he paper is motivated by is the presence of possible epistasis that the conventional methods are missing. Showing such results helps the paper excel the standard methods beyond the accuracy metrics. This is sth that the authors mention in the last sentence of the paper but I think can be extended more.

questions:

The algorithm needs the information of the phylogenetic tree as input. How easy is it to construct the tree? and how stable is the proposed generative model to the possible variations in the construction of the phylogenetic tree?





**Summary Of The Paper:**

The paper develops a new generative model for sequences using Variational AutoEncoders (VAEs) with the key difference that the latent dimension is modeled as a tree-structured Ornstein-Uhlenbeck (OU) process which captures the phylogenetic tree of the sequences.

**Summary Of The Review:**

Overall, the paper tries to address an important and interesting problem in learning generative models of homologous sequences informed by their phylogenetic structure. However, the results in particular are not convincing enough to grant acceptance. The paper can be improved by adding evidences of some biological insights gleaned from the model re the evolutionary process behind the sequence. Alternative the paper can focus on the computation aspect and expand on the reasons behind scaling and the extend it can tolerate larger data sets.

---

> ### Author Response · Authors · 2021-11-19
> **Author's comment**
>
> As we point out in the answer to referee 1Min, our aim was not to claim that Draupnir outperforms ASR programs. Rather, we show (for the first time) how evolutionary information can be incorporated in a deep generative model that goes beyond factorized evolution. We believe this is amply demonstrated by the quantitative ASR results, the more qualitative ablation studies and a newly added evaluation concerning co-evolution (see point 3 in the answer to referee 1Min, which also addresses questions concerning the presence of epistasis of this referee). Claiming superiority to generative models that are geared to big data or ASR methods that have been optimized for many years would be preliminary. We have also added a more quantitative analysis of the results in Fig. 7.
>
> The standard deviations of the average accuracy results shown in Fig.3 are included in the appendix (Tables 5 and 6).
>
> Using an existing tree (which is easy to obtain) is a standard practice in ASR (Joy et al, 2016). Moreover, a study (https://www.cs.uoregon.edu/Reports/DRP-200903-Hanson-Smith.pdf) by the University of Oregon of the influence of phylogenetic uncertainty on ASR concluded:
>
> "My results are surprising and nonintuitive: phylogenetic uncertainty is not correlated with the accuracy of reconstructed ancestral states. The conditions which produce phylogenetic uncertainty result in ancestral states on alternate trees which are similar, if not identical, to the ancestral states on the maximum likelihood tree. Ultimately, integrating phylogenetic uncertainty does not significantly affect the accuracy of reconstructed ancestral sequences."
>
> Nonetheless, the case of a latent tree is most definitely interesting, though clearly beyond the scope of the current study and rebuttal period. We believe the brief discussion given at the end of the article points to a realistic and feasible way forward.

---

> ### Comment · Area_Chair_BL2M · 2021-11-24
> **Respond to author feedback**
>
> Please respond to author feedback and other reviewers' comments and indicate if it changes your rating.

---

### Official Review · Reviewer_1Min · 2021-11-03

**Correctness:** 3
**Technical Novelty And Significance:** 4
**Empirical Novelty And Significance:** 2
**Recommendation:** 8
**Confidence:** 4

**Main Review:**

Overall this is an exciting paper and the field should welcome more work along these lines. The authors have put a lot of work into developing their model and the presentation is quite clear. My main feedback is concerned with the model evaluations and how they relate to overall claims of the paper.

Strengths:
- The technical novelty of relaxing independent sites is interesting and important. Careful understanding of when it is useful to relax this assumption and how to do it best could yield scientific insights and improve models.
- The use of a tree-structured OU process over latent space is novel and natural for this problem setting.
- The authors compare to a range of baselines that seem relevant and appear to have run baselines correctly (although this is important to confirm).
- The exposition of the model itself is easy to follow and well-written.

Weaknesses:
The main weakness of the paper is that the evaluations are not systematic enough to back up certain core claims of the paper. Please note that some of the suggestions I make below are *not* feasible in a rebuttal period, so I offer them for consideration in followup work or future iterations.

Scale - The authors claim in the abstract that their method "scale[s] to larger data sets." I see a few key issues here
1. The authors do not explain why they can't run methods on larger trees. They simply say "most ASR methods stop working for data sets with more than 200 leaves." Is this because of excessive compute time, memory constraints, software failing, etc.? I can't find any information on hardware used for these experiments either, which is essential to claims about scale.
2. FastML is the best-performing method for nearly all instances where it's run. The authors should subsample large trees to 200 leaves and compare their method trained on the whole tree to FastML on a subset of the tree. The algorithm "prune down over-represented clades and run FastML" is a reasonable one for trees with lots of leaves and an important baseline.

Representation Quality - In the abstract the authors state that "Our results and ablation studies indicate that the explicit representation of evolution using a suitable tree-structured prior has the potential to improve representation learning of biological sequences considerably"
1. The main evidence I see for this claim is the tSNE clustering plots in figures 2 and 7. These plots are excellent evidence that the tree OU is implemented well, but this evidence alone doesn't suggest major potential for improvement.
2. A direct comparison to other representation learning methods would ground this claim substantially. Without any comparison to other sequence representations, it's hard to assess. For example, the set of MSAs + DMS studies used by DeepSequence could be turned into trees + DMS studies. Then the correlation of DeepSequence ELBOs and Draupnir ELBOs to fitness could be compared.

Independent Sites - The authors state in Section 2.2 "The aim of this work is to go beyond the assumption of independent, factorized evolution." This is an exciting aim but I feel that it needs to be evaluated more carefully.
1. The authors should provide an ablation study of their sequence likelihood where they use a position-wise MLP instead of a GRU. The ASR performance of this model to Draupnir would give a sense for the role of independent sites assumptions in performance. (Note: I do not believe that model is an independent sites model because you still integrate out the latent z's)
2. For the benchmarks offered, we are given no sense of how violated independent sites is in these protein families. One very helpful baseline would be comparing FastML ASR performance on a full tree versus a tree where known coevolving positions are filtered. Showing that Draupnir has a smaller delta in this setting would provide clearer evidence that independent sites is being relaxed in a useful way.

General Suggestion: Something to consider for future work. It is common for papers that train VAEs on MSAs to claim they have implicitly learned phylogenetic signal in a protein family. This paper has an excellent opportunity to evaluate that claim by comparing models which do not use phylogeny to one which explicitly does!

Things that would improve my score:
- Reporting results with sequence likelihoods that factorize over positions
- Running FastML on filtered versions of large trees for comparison
- Direct comparison of Draupnir to other representations of protein or DNA sequences
- Taking a family where independent sites is known to be violated and doing a case study on baselines vs Draupnir

**Summary Of The Paper:**

The authors introduce a VAE for modeling individual protein families that incorporates phylogenetic trees through an OU process on latent space. They also use a sequence likelihood which does not factorize over positions. The authors claim these two advances represent a more expressive and efficient model of protein evolution and apply it to ancestral sequence reconstruction.

**Summary Of The Review:**

This paper is timely and addressing a very important area of work for the protein-ml field. The model is thoughtfully implemented and presented, but the evaluations do not enable strong conclusions to be reached yet. More careful ablations and comparisons are needed to understand implications of this work for scalability, representation learning, and evolutionary modeling.

---

> ### Author Response · Authors · 2021-11-19
> **Author's reply**
>
> 1. On scaling to larger data sets
>
> ASR algorithms programs vary in the use of evolutionary models, the strategy used to reach the optimal solution (marginal reconstruction, joint reconstruction,...) and so on, resulting in widely different demands with respect to memory and computational performance (Joy et al, 2016). The FastML server (http://fastml.tau.ac.il/), for example, limits the amount of sequences to 200.
>
> All our tests were run on the same hardware (Intel(R) Xeon(R) Gold 6136 CPU @ 3.00GHz, Quadro RTX 6000 GPU). None of the ASR programs (as far as we know) can make use of the GPU, though they make use of multi-threading on the CPU. We used recommended default parameters for the ASR programs (which we now report in the Appendix).
>
> We agree that the claim "scales to larger data sets" is misleading, as running the ASR programs on more powerful hardware (or subsampling large trees) might well produce good results. Therefore, we removed such unwarranted claims from the article.
>
> 2. On representation quality and direct comparison to other representation learning methods.
>
> The aim of the article is not to claim that our evolutionary approach at this point (a) leads to better representations compared to state-of-the-art VAE methods estimated using big data or (b) outperforms ASR programs that have been optimized for many years over many data sets. Rather, we show (for the first time) how evolutionary information can be incorporated in a deep generative model using a probabilistic, non-factorized model of evolution and used for a classic problem in protein bioinformatics: ancestral sequence reconstruction. We believe this is amply demonstrated by the quantitative ASR results, the ablation studies (we provide a more quantitative evaluation of the results shown in Fig 7) and a newly added section that evaluates if the model can indeed go beyond factorized evolution (see next point). We have removed any sentences that seem to claim (a) or (b).
>
> 3. On going beyond factorized models of evolution:
>
> We agree that our claim "to go beyond factorized evolution" needs to be evaluated more carefully. Therefore, we added a section (see Appendix) that evaluates the Direct Information values (DI) from a DCA analysis over all positions pairs in sequences sampled at the root node for a protein family from a variational posterior using an amortized guide. The results show that the variational posterior indeed seems to be capable to capture co-evolution compared to a factorized baseline and the co-evolutionary signal present in the leaf sequences.

---

> > ### Comment · Reviewer_1Min · 2021-11-29
> > **Thank you! Score increased**
> >
> > The reviewers addressed all of my questions about scope of claims and how they contextualize their work within the broader literature. Additionally they added an interesting experiment comparing DCA on a real MSA to DCA on an MSA sampled from evolution captured by Draupnir.
> >
> > With these updates, the core claim of the paper is that they introduce a deep generative model of protein evolution which achieves competitive performance to state of the art ASR methods, provide evidence that it captures coevolution, and has potential to scale well without use of high-performance computing hardware. This is an excellent paper that is in scope for ICLR and I am upgrading my score from a 5 to an 8.

---

> ### Comment · Area_Chair_BL2M · 2021-11-24
> **Respond to author feedback**
>
> Please respond to author feedback and other reviewers' comments and indicate if it changes your rating.

---

### Decision · Program_Chairs · 2022-01-20

**Decision:**

Accept (Poster)

**Comment:**

The paper describes a VAE model for individual protein families that uses phylogenetic trees through an Ornstein-Uhlenbeck process on latent space. They also use a sequence likelihood which does not factorize over positions. The paper claims these two advances represent a more expressive and efficient model of protein evolution and apply it to ancestral sequence reconstruction.

Strengths:

- The technical novelty of relaxing independent sites is interesting and important
- The use of a tree-structured OU process over latent space is novel and natural for this problem setting
- The exposition of the model itself is easy to follow and well-written
- A statistically well grounded approach

Weaknesses:

- The evaluations do not enable strong conclusions to be reached yet. More careful ablations and comparisons are needed to understand implications of this work for scalability, representation learning, and evolutionary modeling
- Lack of computational cost details

A majority of reviewers voted with high confidence for acceptance. The only reviewer voting for rejection did not respond to the author's rebuttal and did not give strong arguments for rejection. The authors are encouraged to address the reviwers' comments and improve the paper for the camera ready version.